# Distribution of Bacterial Endosymbionts of the *Cardinium* Clade in Plant-Parasitic Nematodes

**DOI:** 10.3390/ijms24032905

**Published:** 2023-02-02

**Authors:** Sergey V. Tarlachkov, Boris D. Efeykin, Pablo Castillo, Lyudmila I. Evtushenko, Sergei A. Subbotin

**Affiliations:** 1All-Russian Collection of Microorganisms (VKM), G.K. Skryabin Institute of Biochemistry and Physiology of Microorganisms, Pushchino Scientific Center for Biological Research of the Russian Academy of Sciences, Pushchino 142290, Russia; 2Center of Parasitology of A.N. Severtsov Institute of Ecology and Evolution of the Russian Academy of Sciences, Leninskii Prospect 33, Moscow 117071, Russia; 3Institute for Sustainable Agriculture (IAS), Spanish National Research Council (CSIC), Avenida Menéndez Pidal s/n, Campus de Excelencia Internacional Agroalimentario, ceiA3, 14004 Córdoba, Spain; 4California Department of Food and Agriculture, Plant Pest Diagnostic Center, Sacramento, CA 95832, USA; 5Department of Entomology and Nematology, Hutchison Hall, University of California, Davis, CA 95616, USA

**Keywords:** *Cardinium*, genome, nematode, novel group, *Paenicardinium*, phylogeny

## Abstract

Bacteria of the genus “*Candidatus* Cardinium” and related organisms composing the *Cardinium* clade are intracellular endosymbionts frequently occurring in several arthropod groups, freshwater mussels and plant-parasitic nematodes. Phylogenetic analyses based on two gene sequences (16S rRNA and *gyrB*) showed that the *Cardinium* clade comprised at least five groups: A, B, C, D and E. In this study, a screening of 142 samples of plant-parasitic nematodes belonging to 93 species from 12 families and two orders using PCR with specific primers and sequencing, revealed bacteria of *Cardinium* clade in 14 nematode samples belonging to 12 species of cyst nematodes of the family Heteroderidae. Furthermore, in this study, the genome of the *Cardinium* cHhum from the hop cyst nematode, *Heterodera humuli,* was also amplified, sequenced and analyzed. The comparisons of the average nucleotide identity (ANI) and digital DNA–DNA hybridization (dDDH) values for the strain *Cardinium* cHhum with regard to related organisms with available genomes, combined with the data on 16S rRNA and *gyrB* gene sequence identities, showed that this strain represents a new candidate species within the genus “*Candidatus* Paenicardinium”. The phylogenetic position of endosymbionts of the *Cardinium* clade detected in nematode hosts was also compared to known representatives of this clade from other metazoans. Phylogenetic reconstructions based on analysis of 16S rRNA, *gyrB, sufB, gloEL, fusA, infB* genes and genomes and estimates of genetic distances both indicate that the endosymbiont of the root-lesion nematode *Pratylenchus penetrans* represented a separate lineage and is designated herein as a new group F. The phylogenetic analysis also confirmed that endosymbionts of ostracods represent the novel group G. Evolutionary relationships of bacterial endosymbionts of the *Cardinium* clade within invertebrates are presented and discussed.

## 1. Introduction

The genus “*Candidatus* Cardinium” (Bacteroidetes) was proposed for intracellular endosymbiotic bacteria [1] which were first discovered in deer tick *Ixodes scapularis* [2], and then in mite *Brevipalpus phoenicis* [3] and *Encarsia* wasps [1]. Since then, members of *Cardinium* phylogenetic clade comprising both “*Candidatus* Cardinium” sensu stricto and the related bacteria found in several arthropod groups, freshwater mussels and plant-parasitic nematodes have often been referred to as *Cardinium* or *Cardinium* bacteria for convenience. Currently, there is an estimated 6 to 23% or more of tested arthropod species, which carry endosymbionts from the *Cardinium* clade. However, its incidence is rather heterogeneous among arthropod groups [3,4,5,6,7,8,9,10,11].

The *Cardinium* and *Cardinium*-related symbionts have a wide variety of impacts on host individuals, including feminization, parthenogenesis induction, and cytoplasmic incompatibility [1,2,12]. These may also be mutualistic endosymbionts and may have a potential role in wasp nutrition [13].

Phylogenetic analyses based on two gene sequences (16S rRNA and *gyrB*) showed that the *Cardinium* clade comprised at least five groups. Group A, which includes the genus “*Candidatus* Cardinium” sensu stricto and related organisms, is the largest and the most studied group; members of this group have been found in various arthropod groups: hymenopteran insects, hemipteran insects, mites, opilionids and spiders. This group comprises several putative species, if the similarity values of 16S rRNA gene sequences reported by Noel and Atibalentja [14] and Nakamura et al. [8] are to be considered; these are clearly below the prokaryote species threshold, 98.65% [15]. Group B is noted in plant-parasitic nematodes and includes “*Candidatus* Paenicardinium endonii” described for endosymbionts of the plant-parasitic soybean cyst nematode *Heterodera glycines* [14], and the group C is observed in the biting midges species of the genus *Culicoides* [8]. The above three groups were proposed to integrate into a single species “*Candidatus* Cardinium hertii” [8]. However, low 16S rRNA gene identities (reportedly 92.8–96.3%) [8,14] between members of these groups are indicative of the presence here of at least three genera taking into consideration the recommended genus borderline (94.5% or 95.0%) and current trend in prokaryote systematics to divide polyphyletic genera into monophyletic ones [16,17,18,19]. Subsequently, group D (93–94% of 16S rRNA sequence identity to the closest sister clade, group C) was reported in a single marine copepod species, *Nitocra spinipes,* and presumed to occur in other copepods, while a representative of group E was found in a single mite species, *Achipteria coleoptrata* [20,21,22].

Together with the aforementioned *H. glycines* [14], intracellular symbionts from the *Cardinium* phylogenetic clade were discovered by molecular methods in some other plant-parasitic nematodes, such as the Chinese cereal cyst nematode, *H. sturhani* [23], the root-lesion nematode, *P. penetrans* from the USA [24,25,26] and the spiral nematode *Rotylenchus zhongshanensis* from China [27]. It has been suggested that some intracellular bacterium-like (*Rickettsia*-like) organisms revealed in early studies by transmission electron and light microscopy may belong to the *Cardinium* clade as well. These include symbionts of the potato cyst nematode *Globodera rostochiensis* from Bolivia [28,29,30], the pea cyst nematode *Heterodera goettingiana* from England [28], and the soybean cyst nematode *H. glycines* from the USA [31], as well as the bacterium recently detected in *Bursaphelenchus mucronatus* [32]. Up until now, there have been no published survey investigations on the occurrence of bacterial endosymbionts of the *Cardinium* clade within a wide range of plant-parasitic nematode species.

In the present study, polymerase chain reaction (PCR) primers designed to amplify the *Cardinium* 16S ribosomal RNA gene of *Cardinium* and *Cardinium*-related symbionts were used to screen a large collection of plant-parasitic nematodes in order to estimate the distribution of that symbiotic bacterium and to determine the limits of its host range. We also present an annotated draft genome for a new strain of the *Cardinium* clade recovered from the hop cyst nematode, *Heterodera humuli*, and provide results of comprehensive phylogenetic analyses of the *Cardinium* clade, while erecting new molecular groups within it.

## 2. Results

### 2.1. Distribution of Cardinium and Cardinium-Related Endosymbionts in Plant-Parasitic Nematodes

In order to estimate the infection rate by endosymbionts of the *Cardinium* clade and explore their host range and distribution, DNA extracted from plant-parasitic nematode species was screened using primers designed to amplify a portion of the *Cardinium* 16S ribosomal rRNA gene. Of 142 screened DNA nematode samples belonging to 93 species from 12 families of two orders of plant-parasitic nematodes, a PCR product of expected length of 482 bp was observed from 14 samples belonging to 12 species of cyst nematodes of the family Heteroderidae (Table 1 and Appendix A). A total of 81 samples of cyst nematodes belonging to 39 species of the subfamilies Heteroderinae and Punctoderinae were tested. Endosymbionts were found in four genera of this family: *Cactodera* (two species)*, Heterodera* (eight species)*, Globodera* (one species) and *Punctodera* (one species) (Table 1). *Cardinium* was detected in only 1/4 populations of *H. avenae,* 2/3 of *H. latipons*, 2/9 of *H. mediterranea*, 1/2 of *H. salixophila* and 1/10 of *G. rostochiensis* (Appendix A). *Cardinium*-specific *gyrB* primers were only applied for nematode samples, which showed positive results with *Cardinium*-specific 16S rRNA primers. PCR products and sequences of the *gyrB* gene fragment were obtained from 13 samples (Table 1).

### 2.2. Phylogenetic Relationships among Representatives of Cardinium Clade from Plant-Parasitic Nematodes and Other Host Organisms

Relationships of 16S rRNA gene sequences obtained from plant-parasitic nematodes are given in Figure 1A. Some relationships are not well supported because of a low informative signal in the gene fragment. Relationships within the *Cardinium* clade based on the *gyrB* gene fragment sequences are presented in Figure 1B.

Several methods have been applied to study relationships within the *Cardinium* clade: (*i*) statistical parsimony analysis of short fragments of 16S rRNA gene sequences available in the GenBank (Appendix A); (*ii*) Bayesian inference and maximum likelihood analysis of nucleotide sequences of 16S rRNA (Figure 2), *gyrB* (Figure 3A), *sufB* (Appendix A), *gloEL* (Appendix A), *fusA* (Appendix A), and *infB* (Appendix A) genes; (*iii*) Bayesian inference analysis of amino acid sequences of *gyrB* (Figure 3B), *sufB* (Appendix A), *gloEL* (Appendix A), *fusA* (Appendix A), and *infB* (Appendix A) genes from selected organisms belonging to different groups and (*iv*) distance-based analysis of genome assemblies (Appendix A).

A phylogenetic network for the 16S rRNA gene sequences (432 sequences, alignment length-461 bp) reconstructed using statistical parsimony (SP) and a tree derived from these gene sequences (27 sequences, 1464 bp) reconstructed using BI and ML are given in Appendix A and Figure 2, respectively. These methods allowed the discrimination of five known (A–E) and two new (F, G) groups within the *Cardinium* clade. The new group F included the endosymbiont from the root-lesion nematode of *P. penetrans,* and the new group G included bacteria from ostracods. In *Cardinium*, the largest substitution differences in the 16S rRNA gene were between *Cardinium* of *H. glycines* (1448 bp) and bacteria of *Nitocra spinipes* (855 bp)-7.7% and between *Cardinium* of *H. humuli* (1448 bp) and bacteria of *N. spinipes*-7.3%. The sequence of the new group F differed between 4.6 and 4.8% from those of group B. The intragroup variation for group B was 4.2%. The smallest difference of the group F sequence was 3.5% with that of *Cardinium* of *Tetranychus urticae*. The 16S rRNA gene sequence of the new group G differed by 5.3–7.2% from sequences of other groups.

Phylogenetic relationships within the *Cardinium* clade as inferred from the analyses of *gyrB*, *sufB*, *gloEL*, *fusA*, *infB* genes are given in Figure 3, Appendix A. Sequences of *Cardinium* of *P. penetrans* designated as the molecular group F did not form a clade with sequences of *Cardinium* of *Heterodera* spp. in the majority of trees, except for the nucleotide *gloEL* and amino acid *infB* gene trees.

The phylogenomic tree inferred from 15 bacterial genomes of the *Cardinium* clade using JolyTree is given in Appendix A. *Cardinium* of group A was monophyletic, while *Cardinium* of *P. penetrans* formed a separated lineage and *Cardinium* of *H. glycines* and *H. humuli* had a sister relationship.

### 2.3. Genome Features of Endosymbiont of Heterodera humuli

Genome sequencing and assembly details are listed in Table 2 and Appendix A. Data are deposited in NCBI with the accession number JAOPFT000000000.1. Draft genome assembled form 3,523,302 paired end reads of 150 bp length, obtained from Illumina NovaSeq. The assembly consisted of 141 scaffolds with an N50 of 83,838 bp, a longest scaffold of 244,505 bp, and a total length of 1,056,324 bp. Genome coverage was 452. This genome revealed 38.55% G+C, 935 predicted proteins, 3 rRNA, 36 tRNA genes and three ncRNAs, with 80.8% of the genome comprising coding regions and 29.8% of predicted proteins having no known function. There was no evidence of a plasmid. There were at least two variants of *gyrB* and *gloEL* gene sequences for the endosymbiont of *H. humuli*, and these variants have 0.1% and 0.4% nucleotide differences, respectively, and differed in one amino acid. Genome details compared to two other members of the *Cardinium* clade are listed in Table 2.

In general, genomes of the above three strains have similar characteristics (Table 2). Only slight variations were observed; for instance, the cHhum genome is the smallest while having the largest G + C content. Among these genomes (Appendix A), greater orthogroups overlap was observed between cHhum and cHgTN10 (649 shared orthogroups) than between cHhum and cPpe (500 shared orthogroups); overlap between all genomes comprises 491 orthogroups. Core overlap consists of 1654 genes across all three genomes; 13% genes had an unknown function. At the same time, 85 orthogroups (100 genes in total) are unique for the cHhum genome and contain 61% genes with an unknown function.

The values of average nucleotide identity (ANI) and digital DNA-DNA hybridization (dDDH) based on a genome-to-genome sequence comparison calculated for strain *Cardinium* cHhum from the hop cyst nematode, *Heterodera humuli,* with regard to related organisms with available genomes are shown in Appendix A. The values determined were in the range of 68.90–93.96% (ANI) and 19–55% (dDDH), with the highest values revealed between *Cardinium* cHhum and *Cardinium* cHgTN10 (93.96% and 55%, respectively). These are clearly below 95–96% (ANI) and 70% (DDH) used as boundaries for prokaryote species delineation [33,34]. No data were reported on the genome sequence for the type strain of “*Candidatus* Paenicardinium endonii” at the time of its description. However, the high identities of 16S rRNA and *gyrB* genes of *Cardinium* cHgTN10 and the above type strain (100% and >99.5%, respectively) together with the same nematode host species of these strains, indicate that *Cardinium* cHgTN10 belongs to the species “*Candidatus* Paenicardinium endonii”, and therefore *Cardinium* cHhum from *H. humuli* represents a new candidate species in the genus “*Candidatus* Paenicardinium”.

**Table 2 ijms-24-02905-t002:** Assembly details and genome features for strains of *Cardinium* clade; endosymbionts of plant-parasitic nematodes.

Characteristics	*Cardinium* cHhum	*Cardinium* cHgTN10	*Cardinium* cPpe
GenBank accession no.	JAOPFT000000000.1	CP029619.1	RARA00000000.1
Reference	This study	Showmaker et al. [35]	Brown et al. [25]
Host	*Heterodera humuli*	*Heterodera glycines*	*Pratylenchus penetrans*
Coverage (×)	452	30	15
No. of scaffolds	141	- ^†^	27
Scaffold N50 (bp)	83,838	- ^†^	163,560
Assembly size (bp)	1,056,324	1,193,042	1,358,212
G+C content (%)	38.6	38.2	35.8
No. of protein-coding genes	935	943 ^‡^	1075 ^‡^
No. of hypothetical proteins	279	263 ^‡^	255 ^‡^
No. of rRNAs	3	3 ^‡^	3 ^‡^
No. of tRNAs	36	37 ^‡^	38 ^‡^
No. of ncRNAs	3	2 ^‡^	3 ^‡^
No. of pseudo genes	30	34 ^‡^	115 ^‡^
Coding regions length (%)	80.8	81.1 ^‡^	76.9 ^‡^

^†^ Complete genome. ^‡^ RefSeq annotation.

## 3. Discussion

It has been shown by many studies that some taxonomic groups, such as parasitic wasps, mites, and spiders, show high frequencies of infection by the bacterial endosymbionts of the *Cardinium* clade, while the majority of arthropod groups are either free of or seldom harbor these bacteria [36]. Our survey of 93 plant-parasitic species revealed that endosymbionts of this phylogenetic group are widespread only in cyst-forming nematodes of the family Heteroderidae. These occurred in 12 (30%) of the 39 tested species of cyst-forming nematodes. Our results confirmed the presence of these bacteria in the potato cyst nematode, *Globodera rostochiensis* from Bolivia and the pea cyst nematode, *H. goettingiana*, in which this bacterium has been previously detected by microscopic method only [28], and also revealed *Cardinium* in *Cactodera rosae*, *Cactodera* sp., *H. humuli*, *H. ripae*, *H. salixophila*, *H. latipons* and *Punctodera chalcoensis*, in which this intracellular endosymbiont has not been reported earlier. Discovery of two variants of *gyrB* and *gloEL* gene sequences of *Cardinium* cHhum may indicate the presence of two different functional copies of the above genes in this bacterium or the presence of two strains of this bacterium in *H*. *humuli.*

It has been revealed in several studies that lateral transfer of *Wolbachia* genome fragments into the host nuclear genome has occurred in arthropods and nematodes that carry live infections [37,38,39]. In one case, the inserted *Wolbachia* genes were transcribed within eukaryotic cells lacking an endosymbiont [38]. The possibility cannot be excluded that lateral transfer of *Cardinium* genome fragments could also occur into the nematode nuclear genome, that might potentially lead to a false positive detection result of the presence of alive symbionts in a host using a species-specific PCR approach. It has been also shown that the majority of the potential protein-coding genes in the *Wolbachia*-like fragments in host genomes contained many insertions, deletions, frameshift mutations or nonsense codons compared to their homologues from living *Wolbachia* genomes [37,38]. Considering the fact that our sequence analysis showed the presence of functional copies of the *gyrB* gene obtained in the results of amplification and the fact of previous microscopic detections of intracellular bacterium-like in cyst nematodes [28,29,30,31], we believe that live infections and not cases of palaeosymbiosis have been detected in our samples. FISH (fluorescence in-situ hybridization) might be done to reveal locations of endosymbionts within nematode bodies in future studies.

Comparison of phylogenetic relationships of endosymbionts from the *Cardinium* clade found in cyst-forming nematodes using 16S rRNA and *gyrB* genes with phylogeny of the nematodes reconstructed based on ITS rRNA, 28S rRNA and other nematode genes [40] did not reveal strict co-phylogenetic patterns that would indicate a possible horizontal transmission of these symbionts among cyst-forming nematodes, and which could occur among species inhabiting similar localities. Phylogenetic analysis revealed that endosymbionts of cyst-forming nematodes belonging to the subfamily Punctoderinae formed two separate clades that were likely transferred independently at least twice to this group.

Currently, limited information is available on the effects of endosymbionts from the *Cardinium* clade on plant-parasitic nematodes. Walsh et al. [30] noticed that stored, hatched second-stage juveniles of *G. rostochiensis* appeared to contain more microorganisms than newly hatched juveniles; however, no obvious signs of pathogenicity were observed. Uninfected juveniles apparently lived longer than infected ones in vitro, probably because infected juveniles exhausted their lipid reserves sooner than uninfected juveniles. The study of nematode symbionts is important, as the symbiosis may potentially have an impact on the physiology of a host and this knowledge could be used for nematode control.

Stouthamer et al. [41] noticed the issue of obtaining enough bacterial DNA for sequencing, particularly from hosts that were rather small, because the absolute amount of bacterial DNA per host is at least somewhat proportional to host body size. These authors proposed a protocol designed to enrich endosymbiont DNA in samples, particularly for hosts that were small and/or harbored endosymbionts at relatively low densities. This extraction technique produces a higher yield of endosymbiont-enriched DNA of a quality and length appropriate for long and short read libraries. In our study, we used whole genome amplification of single cysts to enrich DNA, and this procedure allowed us to increase the DNA yield and led to higher read numbers by factors of 20 in sequence datasets.

At the time of writing, whole and draft genomes of nematode endosymbionts from the *Cardinium* phylogenetic group have been assembled and published only for strains from *H. glycines* [35] and *P. penetrans* [25]. In this work we present an annotated draft genome sequence of endosymbionts for a third plant-parasitic nematode species, *H. humuli.* Genome analysis suggested a greater similarity between endosymbionts of two cyst-forming nematodes.

Phylogenetic analyses of members of the *Cardinium* clade based on two gene sequences (16S rRNA and *gyrB*) suggested the existence of at least five groups designated as A, B, C, D and E. Group B originally included “*Candidatus* Paenicardinium endonii” from *H. glycines* [8] and was then supplemented with endosymbiont of *P. penetrans* [25]. However, our analysis of phylogenetic patterns and genetic distances using several genes and genomes revealed that this *P. penetrans* endosymbiont forms a separate lineage (the new molecular group designated F). Group B, on the other hand, includes *Cardinium* cHhum and other nematode symbionts. The comparisons of ANI and dDDH values and the sequences of 16S rRNA gene and the *gyrB* nucleotide and amino acid sequences showed that *Cardinium* cHhum represents a new candidate species within the genus “*Candidatus* Paenicardinium”.

We confirmed the presence of other groups reported (A, B, C, E) and suggest another new group G from ostracods. Schön et al. [42] found evidence for the general presence of *Cardinium* bacteria in natural populations of various non-marine ostracod species. These authors also showed, based on 16S rRNA gene data phylogenetic reconstructions, that bacteria from ostracods represented a novel lineage with a monophyletic origin, but a new group was not erected. Our phylogenetic analysis revealed that group G contains additionally an endosymbiont from a freshwater mussel, *Unio crassus* recently obtained by Mioduchowska et al. [43].

It is worth noting that, in common with group A which includes “*Candidatus* Cardinium” sensu stricto and closely related organisms, and group B with “*Candidatus* Paenicardinium”, other molecular groups appear to comprise organisms of different bacterial genera as well. This can be inferred at least from the 16S rRNA gene sequence similarities between representatives of these groups [8,14,20,22], which are below the commonly accepted genus borderline (94.5% or 95.0%) [16,19]. Only group F (strain *Cardinium* of *Pratylenchus penetrans*) shares higher sequence identities to a few groups (according to our Blast search) and requires further elucidation of its taxonomic status using genome relatedness indices, such as the average amino acid identities (AAI), percentage of conserved proteins (POCP), the average nucleotide identities (ANI) and genome alignment fractions (AF) [18,19].

It has been shown that symbionts from the *Cardinium* clade are maintained within host populations via maternal transmission to the progeny through the egg cytoplasm. Because representatives of various lineages of this group are shared with different and distantly related host species, it was surmised that these endosymbionts may also exchange hosts via horizontal transmission [4,44,45]. It seems that eukaryotic host cells may not require the survival of this bacterium, as it has been shown for *Wolbachia*. The ability of intracellular bacterial endosymbionts to survive outside host cells may increase the probability of successful horizontal transfer and the exploitation of new ecological niches [46].

Phylogenetic relationships and evolution trends within the *Cardinium* clade have been studied and discussed by many authors [4,6,8,11,14,22,25,27,39,41,47,48,49]. Members of the *Cardinium* clade represent the sister group to the amoeba symbiont *Amoebophilus asiaticus* [8,50]. It has been shown that *Cardinium* symbionts share several genome characteristics with the amoeba symbiont *Amoebophilus*. It is thus likely that the common ancestor of the *Cardinium* clade (pro-*Cardinium*) and *Amoebophilus* lived as a symbiont of an amoeba or a protist [13] inhabiting a water environment. Further evolution of endosymbionts occurred in several radial directions with their transition to freshwater ostracods, copepods and mussels, soil-inhabiting oribatid mites, biting midges, whose larvae develop in a variety of semi-aquatic or aquatic habitats, and to soil plant-parasitic nematodes (Figure 4). The most flourishing group of the *Cardinium* clade is group A, which possibly originated from soil mites, where these bacteria were widely circulated. From spider herbivorous mites, which were found to be particularly rich in *Cardinium* infections [36,48], the bacteria have transformed to phloem-feeding insects with piercing-sucking mouthparts, and through them to plants. It has been demonstrated that an interspecific horizontal transmission of symbionts of the *Cardinium* clade can occur through the plant tissue pierced by the insect host, which releases bacteria from its salivary glands during feeding [45,51]. Further evolutionary transition of symbionts occurred from phloem-feeding insects (whiteflies, leafhoppers, scale insects) to endoparasitic wasps and from other insects and mites to its natural enemies, spiders and opilionids. Low sequence divergence between predators and preys may indicate recent occurrences of these transitions. The presence of several lineages of *Cardinium* and relatives from opilionids and spiders [47] indicated independent horizontal movements of this bacterium to these arthropods during evolution.

## 4. Materials and Methods

### 4.1. Nematode Populations

Ninety-three species of plant-parasitic nematodes belonging to several major families of the order Tylenchida and *Bursaphelenchus mucronatus* from the order Aphelenchida collected from various locations (Table 1 and Appendix A) were used for the *Cardinium* screening study. A total of 142 nematode samples belonging to 93 species from 12 families were analyzed. Nematode species were identified using morphology and molecular datasets. Cysts of *H. humuli* and *H. ripae* collected from common nettle (*Urtica dioica* L.) growing on the bank of the Jauza River, Moscow [52], were used for genomic sequencing of *Cardinium.*

### 4.2. DNA Extraction and Whole-Genome Amplification

For the *Cardinium* screening study, DNA samples each obtained from single or several specimens of plant-parasitic nematodes were used. Nematodes were washed in distilled water to exclude possible environmental bacterial contamination and then placed into a drop of water. Nematodes were cut under a binocular microscope using an eye scalpel. Fifteen μL of cut nematode suspension in water were transferred into a 0.2 mL Eppendorf tube; 3 μL proteinase K (600 μg ml^−1^) (Promega, Madison, WI, USA) and 2 μL 10 × PCR buffer (Taq PCR Core Kit, Qiagen, Germantown, MD, USA) were added to each tube. The tubes were incubated at 65 °C (1 h) and 95 °C (15 min) consecutively. After incubation, the tubes were centrifuged and stored at −20 °C until used.

Three DNA samples from cyst nematodes were submitted for the *Cardinium* genomic sequencing study. DNA for the first sample was obtained from a mixture of several dozen cysts of *H. humuli* and *H. ripae* (CD3144) using a MasterPure Complete DNA & RNA purification kit (Lucigen, Middleton, WI, USA) following the manufacturer’s instructions. DNA for the second sample and the third sample was obtained from individual cysts of *H. humuli* (CD3144_Hum) and *H. ripae* (CD3144_Rip), respectively, using the whole genome amplification (WGA) approach. WGA of DNA extracted from individual cysts with proteinase K protocol was performed using an Illustra GenomiPhi V2 DNA amplification kit (Cytiva, Marlborough, MA, USA) following the manufacturer’s instructions. The products were purified using a QIAquick PCR purification kit (Qiagen, Germantown, MD, USA) and submitted for genome sequencing.

### 4.3. PCR and Sequencing of 16S rRNA for Cardinium

PCR was performed on DNA nematode samples using *Cardinium*-specific 16S rRNA gene primers: Car_281_F (5′-GGT AGG GGT TCT TAG TGG AAG-3′) and Car_269_R (5′-TGC TCC CCA CGC TTT CGT G-3′) designed by Brown et al. [25] and *Cardinium*-specific *gyrB* gene primers: gyrb_859F-mod (5′-ATG CAY GTA ACG GGD TTT AAA AG-3′) modified in this study from Stouthamer et al. [53] and gyrb_1498R (5′-CAT AAT YAC AAT TTT ATG GTA MCG-3′) designed in this study. The specificities of *Cardinium*-specific primers were evaluated in silico using corresponding gene sequence alignments for *Cardinium* and related genera and using a Blast search in the Genbank database. Amplification of partial 16S rRNA and *gyrB* genes was performed in a 0.2 mL Eppendorf tube containing 2.5 μL 10 × PCR buffer, 5 μL Q solution, 0.5 μL dNTPs mixture (10 mM each) (*Taq* PCR Core Kit, Qiagen), 0.15 μL of each primer (1.0 μg μL^−1^), 0.1 μL *Taq* Polymerase, 2 μL DNA nematode extract and double distilled water to a final volume of 25 μL.

All PCRs were run with the following thermal profile: an initial denaturation at 94 °C for 4 min, followed by 40 cycles of 1 min at 94 °C, 1 min at 45 °C, and 1 min 30 s at 72 °C, with a final extension at 72 °C for 10 min. The PCR product was run on 1% agarose gel in a TAE buffer. Amplicons were visualized with a GelGreen nucleic acid stain (Biotium, Fremont, CA, USA) under a UV light, purified by using a QIAquick PCR purification kit (Qiagen, Germantown, MD, USA) according to the manufacturer’s instructions and directly sequenced by Genewiz (San Francisco, CA, USA). New sequences were deposited in the GenBank database; accession numbers are given in Table 1 as well as the phylogenetic trees.

The new sequences of *Cardinium* of *H. humuli* for *fusA*, *gloEL*, *sufB* and *infB* genes were obtained from genome sequencing datasets.

### 4.4. Phylogenetic Analysis

The new sequences of *Cardinium* of *H. humuli* for 16S rRNA, *gyrB*, *fusA*, *gloEL*, *sufB* and *infB* genes were aligned using ClustalX 1.83 [54] with their corresponding published gene sequences of selected *Cardinium* [20,22,25,35,55]. Outgroup taxa were chosen based on previously published data [25]. The best-fit models of DNA and protein evolution were obtained using the jModelTest 2.1.10 [56] and ProtTest3.4.2 [57], respectively, with the Akaike Information criterion. Nucleotide and amino acid sequence alignments were analyzed with Bayesian inference (BI) using MrBayes 3.1.2 and MrBayes 3.2.7, respectively [58]. BI analysis was initiated with a random starting tree and was run with four chains for 1.0 × 10^6^ generations for nucleotide sequence alignment and for 1.0 × 10^4^ generation for amino acid sequence alignment. Two runs were performed for each analysis. The Markov chains were sampled at intervals of 100 generations. After discarding 10% and 25% burn-in samples for nucleotide and amino acid alignments, respectively, 50% majority rule consensus trees were generated. Posterior probabilities (PP) in percentages are given on the appropriate clades. Nucleotide sequence alignments were analyzed with maximum likelihood (ML) using PAUP* 4b10 [59]. Bootstrap support (BS) values for ML trees were calculated by a heuristic search from 100 replicates. The 16S rRNA gene sequence alignment was used to construct phylogenetic network estimation using statistical parsimony (SP) as implemented in POPART software (http://popart.otago.ac.nz) [60]. Trees and the network were drawn with Adobe Illustrator v.10. Pairwise divergence between taxa was calculated as the absolute distance value and the percentage of the mean distance, with adjustment for missing data, using PAUP.

### 4.5. Genome Sequencing and Analysis

DNA library construction and sequencing were conducted by Novogene Co., Ltd., (Sacramento, CA, USA) using a NEBNext Ultra II DNA library prep kit from Illumina (New England Biolabs, Inc., Ipswich, MA, USA) following the manufacturer’s recommendations. Pooled DNA libraries were sequenced on an Illumina NovaSeq 6000 instrument to obtain 150-bp paired-end reads for three samples. The quality of raw reads was evaluated using FastQC [61]. A total of sequence datasets comprising ~139, 143, and 141 million reads was generated for the first sample (*H. humuli + H. ripae* cysts; CD3144), the second sample (WGA of *H. humuli*; CD3144_Hum), and the third sample (WGA of *H. ripae*; CD3144_Rip), respectively (Table 2). Quality filtering and adapters removing were processed with the Trimmomatic V0.39 [62]. Clean reads from the CD3144_Hum sample were mapped with a Bowtie2 v2.4.5 [63] to the *Cardinium hertigii* cHgTN10 genome (CP029619.1) as the closest one. The mapped reads were then collected and assembled using SPAdes v3.15.4 [64]. Resulting contigs were extended using Mapsembler v2.2.4 [65] to resolve dead ends in the initial assembly. Reads were then mapped back to extended contigs, collected, and reassembled. Assembly graphs were then manually checked with Bandage v0.8.1 [66] to remove suspicious sequences. Finally, reads were mapped onto final scaffolds of the *Cardinium* cHhum assembly to assess variations in key genes for systematic purposes. Annotation was performed with NCBI PGAP v6.3 [67]. To evaluate the presence of *Cardinium* in samples CD3144_Rip and CD3144, clean reads from the corresponding libraries were mapped onto the final assembly of *Cardinium* cHhum. Mapped reads were collected, counted, and assembled.

Orthogroups between the *Cardinium* cHhum, *Cardinium* cHgTN10 and *Cardinium* cPpe genomes were found using the OrthoFinder 2.5.1 [68]. The genome relatedness indices, viz., the average nucleotide identity (ANI) and digital DNA-DNA hybridization (dDDH) values, were calculated using JSpecies 1.2.1 [34] and GGDC 3.0 [69] tools, respectively. The phylogenomic tree was inferred by the JolyTree 2.1.211019ac [70].

## Figures and Tables

**Figure 1 ijms-24-02905-f001:**
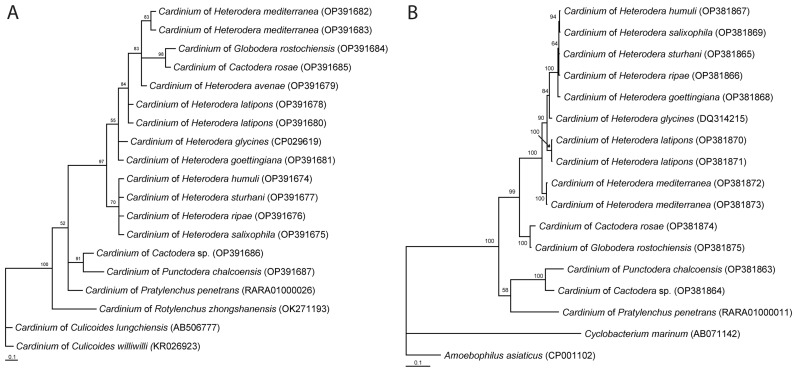
Phylogenetic relationships among representatives of *Cardinium* clade found in plant-parasitic nematodes. Bayesian 50% majority rule consensus tree as inferred from the 16S rRNA gene sequence alignment (alignment length-443 bp) (**A**) and the *gyrB* gene sequence alignment (alignment length-662 bp) (**B**) under the GTR + I + G model. Posterior probabilities (BI) more than 50% are shown at branching points.

**Figure 2 ijms-24-02905-f002:**
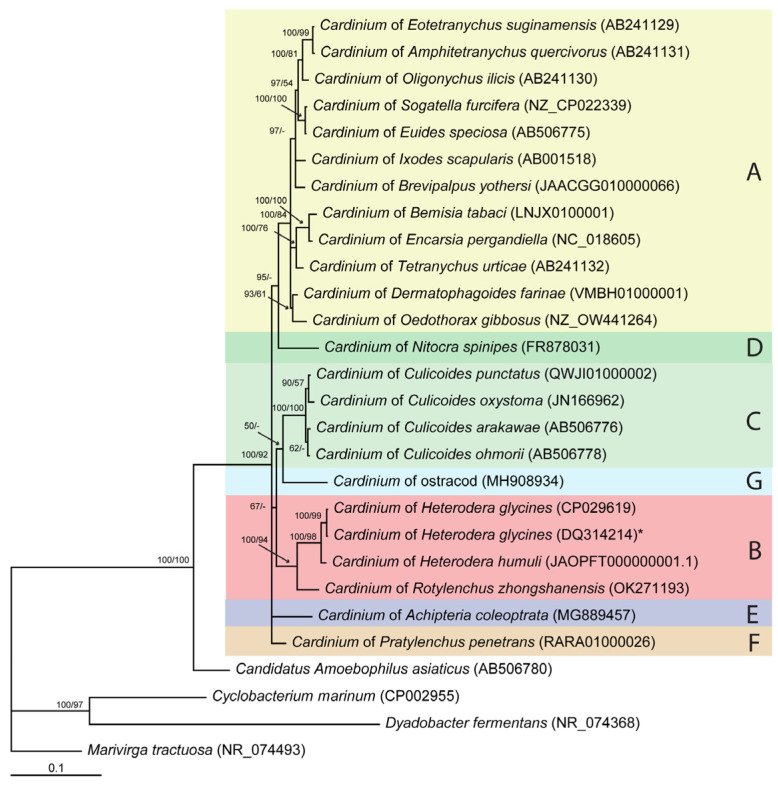
Phylogenetic relationships among representatives of *Cardinium* clade. Bayesian 50% majority rule consensus tree as inferred from 16S rRNA sequence alignment (alignment length-1464 bp) under the GTR + I + G model. Posterior probabilities (BI) and bootstrap values (ML) more than 50% are shown at branching points. *—Type strain of “*Candidatus* Paenicardinium endonii”.

**Figure 3 ijms-24-02905-f003:**
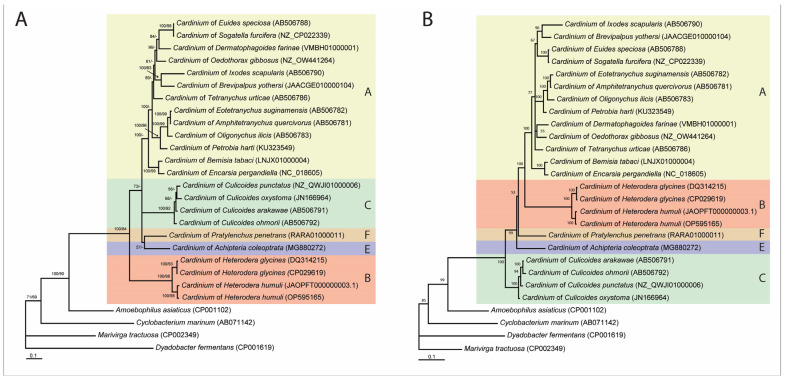
Phylogenetic relationships among representatives of *Cardinium* clade. Bayesian 50% majority rule consensus trees as inferred from *gyrB* nucleotide (alignment length-1414 bp) (**A**) and amino acid (alignment length-470 aa) (**B**) sequence alignments. Posterior probabilities (BI) and bootstrap values (ML) more than 50% are shown at branching points.

**Figure 4 ijms-24-02905-f004:**
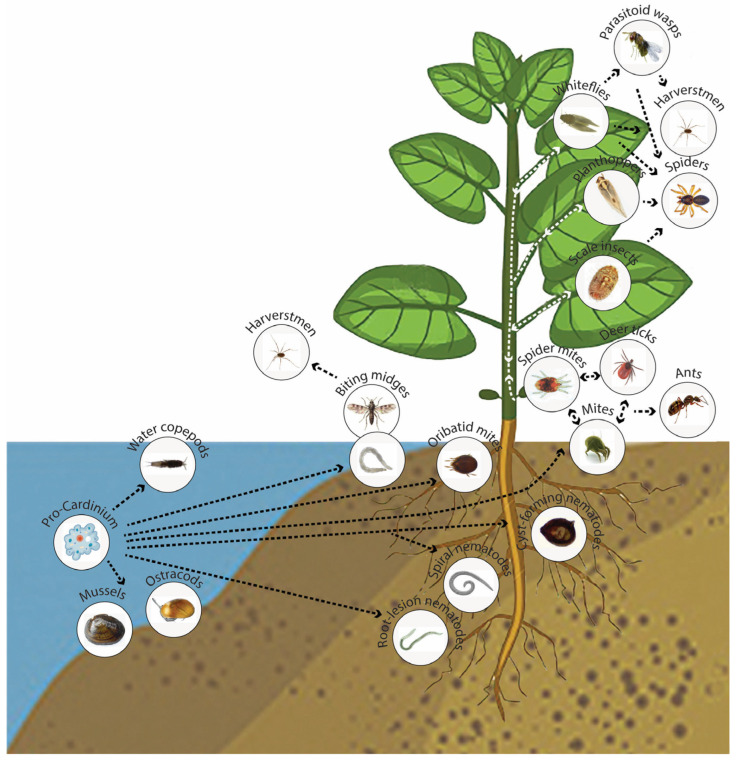
Schema of putative evolutionary relationships and horizontal transmissions of bacterial endosymbionts of the *Cardinium* clade within invertebrates.

**Table 1 ijms-24-02905-t001:** List of cyst-nematode species with *Cardinium* found during this study.

Species	Sample Code	Location	GenBank Accession Number	
			16S rRNA Gene	*gyrB* Gene
*Cactodera rosae*	CD329	Mexico	OP391685	OP381874
*Cactodera* sp.	CD3612	Mexico	OP391686	OP381864
*Globodera rostochiensis*	CD2568b	Bolivia, Cochabamba, Saavedra	OP391684	OP381875
*Heterodera avenae*	CD1456	Tunisia, Siliana	OP391679	-
*H. humuli*	CD3144b-2	Russia, Moscow	OP391674	OP381867
*H. goettingiana*	CD3123a	France, Antibes	OP391681	OP381868
*H. ripae*	CD3144b-4	Russia, Moscow	OP391676	OP381866
*H. salixophila*	CD3258b	Ukraine, Kherson	OP391675	OP381869
*H. latipons*	CD2047	Turkey, Kilis, Merkez	OP391678	OP381871
*H. latipons*	CD1992	Turkey, Elbistan	OP391680	OP381870
*H. mediterranea*	CD3246	Spain, Saler, Valencia	OP391683	OP381873
*H. mediterranea*	CD3118a	Spain, Vejer, Cádiz	OP391682	OP381872
*H. sturhani*	CD2360a	China	OP391677	OP381865
*Punctodera chalcoensis*	CD2813	Mexico	OP391687	OP381863

## Data Availability

Sequencing data are available from the NCBI database, accessions # JAOPFT000000000.1, OP391674-OP391687, OP381863-OP381875, OP595164, OP595165. All other relevant data are included within the manuscript and Appendix A.

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
