# Peer review of "Distribution of Bacterial Endosymbionts of the Cardinium Clade in Plant-Parasitic Nematodes"

_ijms, 2023, doi:10.3390/ijms24032905_

Round 1

Reviewer 1 Report

the article is dealing with biodiversity of endosymbiotic bacteria of the Cardinium clade in the plant parasitic nematodes using molecular approach.

The authors have presented a great piece of work with extensive molecular analyses and there is not much need to be done to improve the quality of the manuscript as it is already well-written. Except for some small changes with regards to the manuscript's language that could be done, the manuscript can be published in this journal.

For details on the minor changes that need to be addressed, please check the attached pdf file.

Author Response

We are grateful for all your comments and suggestions. We accepted all corrections and made corresponding changes in the revised version.

Reviewer 2 Report

Please see attached!

Author Response

We are grateful for all your comments and suggestions.

Abstract:
1) Suggest rephrasing the sentence. Please see below;
Lines 16-18: “Bacteria of the genus “Candidatus Cardinium” and related organisms composing the Cardinium clade are intracellular endosymbionts frequently occurring in various invertebrates including plant-parasitic nematodes”.
There is limited evidence to say that the endosymbiont Cardinium is frequently occurring in invertebrates or nematodes. In fact, Cardinium is not common among invertebrates except for certain arthropods.

Reply: Corrected in Abstract and Introduction.

Introduction:
1) The introduction could be improved by;
i.    adding some more background information 
ii.    identifying the knowledge gaps and importance of the study 
iii.    clearly stated objectives 
iv. improving the flow(cohesion)
For example; authors first discuss incidence of Cardinium in Arthropods (phylum Arthropoda) - I suggest authors to do the same for nematodes (phylum Nematoda) and then for plant-parasitic nematodes, before discussing the impacts of the endosymbiont and the phylogenetic analysis (in general, how common/prevalent is Cardinium in nematodes and in plant-parasitic nematodes? In which genera? Or species? etc.).

Reply: We made some corrections in Introduction, which contains five paragraphs: 1) general information; 2) impact on host; 3) phylogenetic grouping; 4) Cardinium in nematodes including species and genera; 5) research goals. 

.    2)  Lines 84-86: Here, the authors only mention the use of 16S rRNA for the analyses. But in methods section, gyrB gene is also mentioned. It’s not clear which gene primer(s) is/are used for distribution and/or phylogenetic analyses. 

Reply: Corrected. We added two sentences in the end of first paragraph of Results.

.    3)  Suggest rephrasing the following sentences (not clear what’s authors trying to convey) Lines 57-59: “This group comprises several species as follows from the evaluation of similarity values of 16S rRNA gene sequences reported by Noel and Atibalentja [14] and Nakamura et al. [8], which are clearly below the prokaryote species threshold, 98.65% [15]. 
Lines 84-86: “It might be supposed that some intracellular bacterium-like (Rickettsia-like) organisms revealed in early studies by transmission electron and light microscopy belong to the Cardinium clade as well”. 
.    
Reply: Corrected.

Results
1)  Suggest rephrasing the sentence below 
2.1. Distribution of Cardinium and Cardinium-related endosymbionts in plant-parasitic nematodes
Lines 95-97: “To estimate the infection rate by endosymbionts of the Cardinium clade and explore the range of their host range and distribution, DNA extracted from plant- parasitic nematode species were screened using primers designed to amplify a portion of Cardinium 16S ribosomal rRNA gene”. 

Reply: Corrected.

2)  The following is more appropriate for the materials and methods section (not results). Lines 100-102: “A total 100 of 81 samples of cyst nematodes belonging to 39 species of 
the subfamilies Heteroderinae and Punctoderinae was tested”. 
Lines 116-121: “Several methods have been applied to study relationships within the Cardinium
clade: i) statistical parsimony analysis of short fragments of 16S rRNA gene sequences available in the GenBank; ii) Bayesian inference and maximum likelihood analysis of nucleotide sequences of 16S rRNA, gyrB, sufB, gloEL, fusA, infB genes; iii) Bayesian inference analysis of amino acid sequences of gyrB, sufB, gloEL, fusA, infB genes from selected organisms belonging to different groups and iv) distance-based analysis of genome assemblies”. 

Reply: Corrected. We provided additional references to Figures. All this information is presented in Results section.

3)  Table 2. Assembly details and genome features for strains of Cardinium clade, endosymbionts of plant-parasitic nematodes.
Has repetitive rows of information 

Reply: Corrected.

Materials and Methods
1) This section needs more detail. Please see below for suggestions.
4.1. Nematode populations
Lines 320-322: What was the basis for choosing the plant-parasitic nematode species you chose for this study? Is it random, based on availability or considering known phylogenetic relationships among plant-parasitic nematodes? Please specify.

Reply: We analysed representatives of all major plant-parasitic nematode groups. 

4.2. DNA extraction and whole-genome amplification
Lines 3298-329: The authors say “for several specimens of plant-parasitic nematodes, Proteinase K protocol was used”. Does this mean several nematodes per sample were used for DNA extraction (if so, how many?) or several samples (from 142 total) were used for DNA extraction through Proteinase K protocol?” (if so, what about the rest?) - Suggest paraphrasing.

Reply: Corrected.

How many samples were used per species (trials) to detect Cardinium? This is important to eliminate the other possibilities of detecting Cardinium DNA in samples. Please see below

Reply: This information is given in Table S1.

4.4. Phylogenetic analysis
Lines 368-369: Need paraphrasing
“The new sequences of Cardinium of H. humuli for 16S rRNA, gyrB, fusA, gloEL, sufB and infB genes were aligned using ClustalX 1.83 [51] with their corresponding published gene sequences of selected Cardinium [20, 22, 25, 35, 52 and others]”.

Reply: Corrected.

Did you sequence fusA, gloEL, sufB and infB genes as well? Or did you add those sequences to your ClustalX alignments from previously published data? What do you mean by “the new sequences”? Does that indicate “newly sequenced”? How long (bp) are the aligned sequences? The citations must be edited here; what do you mean by “and others”?

Reply: Corrected. Alignment lengths are given in Figure legends and in Result section. Additional information was added to 4.3 – Materials and Method.

2) Other improvements
4.3. PCR and sequencing of 16S rRNA for Cardinium
This information is relevant for the nematode species in which the endosymbiont has not been reported earlier.
16rRNA primers
PCR detects Cardinium DNA in the nematode sample. However, presence of Cardinium DNA in the sample, does not necessarily provide evidence for host to carry live Cardinium.
For example, you can detect Cardinium DNA in nematode samples due to;
1) Palaeosymbiosis (presence of antient horizontally transferred Cardinium DNA fragments in the nematode genome) and not due to actual presence of the endosymbiont.
2) Contamination of samples (acquire Cardinium DNA from the environment)
How would you eliminate these possibilities? The authors may want to consider performing FISH (Fluorescent In-Situ Hybridization) to confirm the endosymbiont presence within nematodes.
The above comment applies to the following statement as well;
Lines 171-172: “There were at least two variants of gyrB and gloEL gene sequences for endosymbiont of H. humuli.

Reply: Corrected. We added additional paragraph to discuss this issue.

gyrB primers
Since these primers were newly designed, how did the authors evaluate the specificity of the primers (how to evaluate non-specific binding of primers to other bacteria). Did you use negative controls (nematodes that do not carry Cardinium) to test the primers?

Reply: Corrected. Additional information was added to 4.3 – Materials and Methods

Round 2

Reviewer 2 Report

The main objective of this study was to screen a collection of plant-parasitic nematodes in order to detect the endosymbiotic bacteria of the Cardinium clade and to estimate their distribution. The authors also present a draft genome of Cardinium cHhum from the hop cyst nematode, Heterodera humuli and reconstructed a phylogeny using 16S rRNA, gyrB, sufB, gloEL, fusA, infB genes.

The overall study was based on PCR to address the main objective. The authors have used two different sets of Cardinium-specific primers for PCR. 

1) 16S rRNA gene primers that have been published in another study

2) gyrB gene primers designed for this study

The authors have only evaluated the specificity of the newly designed primers in-silico. However, the specificity and the efficiency of these newly designed primers need further verification through experimentation.

The authors conclude the presence of the endosymbiont in seven different plant-parasitic nematode species in which the endosymbiont has not been reported earlier (Cactodera rosae, Cactodera sp., H. humuli, H. ripae, H. salixophila, H. latipons and Punctodera chalcoensis). However, they failed to eliminate the other possibilities (described in my previous comments to authors) of finding Cardinium DNA in the nematode samples. i.e., PCR detects Cardinium DNA in the nematode sample. However, presence of Cardinium DNA in the sample, does not necessarily provide evidence for the host to carry Cardinium

Quoting authors, lines 239-241; “Considering the fact that our sequence analysis showed presence of functional copies of gyrB gene obtained in the results of amplification and the facts of previous microscopic detections of intracellular bacterium-like in cyst nematodes, we believe that live infections and not cases of palaeosymbiosis have been detected in our samples.”

Cyst nematodes include variety of species. Previous microscopic detections of intracellular bacteria in one cyst nematode species does not provide evidence for other cyst nematode species to carry the endosymbiont. 

Also, the authors have not provided mRNA evidence to prove actively expressing genes (gyrB). 

They have not eliminated the possibility of environmental contamination of the samples either.

Considering these, the study does not provide enough evidence to support their conclusions. i.e., the study needs further evidence such as microscopic evidence or Fluorescent In-Situ Hybridization evidence to prove the Cardinium presence in the mentioned nematode species.

Author Response

We are grateful for your comments.

1) “The overall study was based on PCR to address the main objective.  The authors have used two different sets of Cardinium-specific primers for PCR.   1) 16S rRNA gene primers that have been published in another study  2) gyrB gene primers designed for this study.  The authors have only evaluated the specificity of the newly designed primers in-silico. However, the specificity and the efficiency of these newly designed primers need further verification through experimentation. “  

We disagree with this statement. We made all experiments confirming specificity of PCR with Cardinium specific primer method. In Materials and Methods, we clearly indicated that we applied PCR with already known Cardinium specific primers for 16S rRNA gene and modified primers for gyrB genes and we gave appropriate references in Materials and Methods. The specificity of these primers was tested in silico and with a wide range of nematode samples in our research (142 samples). Specificity of amplification is also confirmed by the results of sequencing, which showed only sequences of Cardinium genes in samples generated amplicons.

2) “They have not eliminated the possibility of environmental contamination of the samples either. “

We made a corresponding explanation in Materials and Methods indicating that nematodes were washed before DNA extraction. The argument on environmental contamination does not reflect the present knowledge on Cardinium biology. There is no published data indicating that Cardinium sequences could be obtained from environmental samples or can be detected outside the host.

3) “Presence of Cardinium DNA in the sample, does not necessarily provide evidence for the host to carry Cardinium. “

There is no question about the possibility of transferring genes from the bacterial genome to the host genome. However, there was only one publication on bacteria Wolbachia (Dunning Hotopp et al., 2007) with only one case, when Wolbachia genes were transcribed within eukaryotic cells lacking endosymbiont, in all other cases Wolbachia genes were traced together with alive infection for studied host species. We would also like to underline there are not any studies conducted with Cardinium bacteria and showing transfer genes from this bacteria genome to the host genome.

We already addressed this comment and added a large paragraph in Discussion in the revised version.